# Manipulation of Nuclear-Related Pathways During Kaposi’s Sarcoma-Associated Herpesvirus Lytic Replication

**DOI:** 10.3390/v17111427

**Published:** 2025-10-27

**Authors:** Connor Hayward, Katherine L. Harper, Elena M. Harrington, Timothy J. Mottram, Adrian Whitehouse

**Affiliations:** 1School of Molecular and Cellular Biology, Faculty of Biological Sciences, University of Leeds, Leeds LS2 9JT, UK; connor.hayward@pirbright.ac.uk (C.H.); k.l.harper@leeds.ac.uk (K.L.H.); tim.mottram@hotmail.co.uk (T.J.M.); 2Astbury Centre for Structural Molecular Biology, University of Leeds, Leeds LS2 9JT, UK

**Keywords:** KSHV, nuclear architecture, lytic replication, cellular pathways

## Abstract

Kaposi’s sarcoma-associated herpesvirus (KSHV) is the causative agent of Kaposi’s sarcoma (KS) and several lymphoproliferative diseases. As with all herpesviruses, KSHV replicates in a biphasic manner, with the establishment of a latent, persistent infection from which reactivation occurs, resulting in the completion of the temporal lytic replication cycle and production of infectious virions. Herein, we discuss the impact of KSHV lytic replication on the host cell nucleus and nuclear-related pathways. We highlight the dramatic remodelling of the nuclear architecture driven by the formation of viral replication and transcription centres (vRTCs), and the implications for sub-nuclear organelles, and how pathways involved in DNA damage, ribosomal biogenesis and epitranscriptomic regulation are disrupted or modified during KSHV replication. These changes foster an environment favourable for KSHV replication and may provide novel targets and strategies for therapeutic intervention.

## 1. Introduction

Kaposi’s sarcoma-associated herpesvirus (KSHV) is associated with the development of Kaposi’s sarcoma (KS) and several other lymphoproliferative diseases, including primary effusion lymphoma (PEL) and some forms of multicentric Castleman’s disease [1]. Like all herpesviruses, KSHV exhibits a biphasic life cycle consisting of latent persistence and lytic replication [2]. Latency is established in B cells and in the tumour setting, where viral gene expression is limited to the latency-associated nuclear antigen (LANA), viral FLICE inhibitory protein, viral cyclin, kaposins and several virally encoded miRNAs [3,4,5]. Upon reactivation through certain stimuli such as cellular stress, KSHV enters the lytic replication phase, leading to the highly orchestrated and temporal expression of more than 80 viral proteins that are sufficient to produce infectious virions [6,7]. Herein, we highlight the impact KSHV lytic replication has upon nuclear architecture remodelling and the utilisation of multiple nuclear-related host pathways and molecular machinery to enhance virus replication.

## 2. What Impact Does KSHV Lytic Replication Have on the Nuclear Architecture?

### 2.1. Formation of KSHV Replication Centres

KSHV commandeers the nuclear space for its site of replication by assembling viral replication and transcription centres (vRTCs), which enable viral transcription, DNA replication and capsid assembly to all occur in a specialised environment [8]. During the early stages of KSHV lytic replication, viral transcription of early genes and viral DNA replication take place in small RTCs that generally concentrate at the nuclear periphery [9,10,11]. As infection progresses, the nuclear architecture undergoes a striking re-organisation to facilitate viral replication (Figure 1A,B). Small RTCs coalesce into single large globular or kidney-shaped structures that ultimately fill most of the nuclear space, compressing and marginalising the cellular chromatin to the nuclear periphery [8]. Mass spectrometry of vRTCs shows an enrichment of processing, splicing and DNA replication proteins from the host cell, as well as various heatshock proteins [8]. Notably, depletion or inhibition of heatshock protein 70 results in reduced vRTC formation, suggesting a role for the chaperone in scaffolding replication centres and prompting the possibility of repurposing heatshock protein inhibitor compounds as antivirals targeting this process [8].

There are 6 core viral proteins required for lytic DNA replication. These include ORF6 (single-stranded DNA binding protein), ORF9 (DNA polymerase), ORF40/1 (primase-associated factor), ORF44 (helicase), ORF56 (primase) and ORF59 (DNA processivity factor) [12]. Surprisingly, even in the absence of a lytic cycle replication origin and any known initiator or origin binding protein, the protein products of these six KSHV core replication genes can cooperate when overexpressed in mammalian cells to form large globular pseudo-replication compartments, pseudo-RCs, which exclude cellular DNA [13]. As mentioned above, vRTCs perform multiple distinct roles during the lytic replication cycle, namely viral transcription, DNA replication, genome packaging and capsid assembly. How these membraneless virus-induced structures perform these varied tasks is unknown, and it may be the case that these distinct functions occur in different areas of the vRTCs, leading to some form of compartmentalisation. However, the mechanism by which this may occur is unknown. Recent advances in the focus ion beam (FIB) milling in combination with cryo-electron tomography imaging have yielded a new understanding with regard to the formation and function of cytosolic replication centres produced by RNA viruses [14,15]. The utilisation of such techniques will prove a challenge given the dense, crowded nature of the nuclear landscape, yet remains an alluring prospect for pursuit.

**Figure 1 viruses-17-01427-f001:**
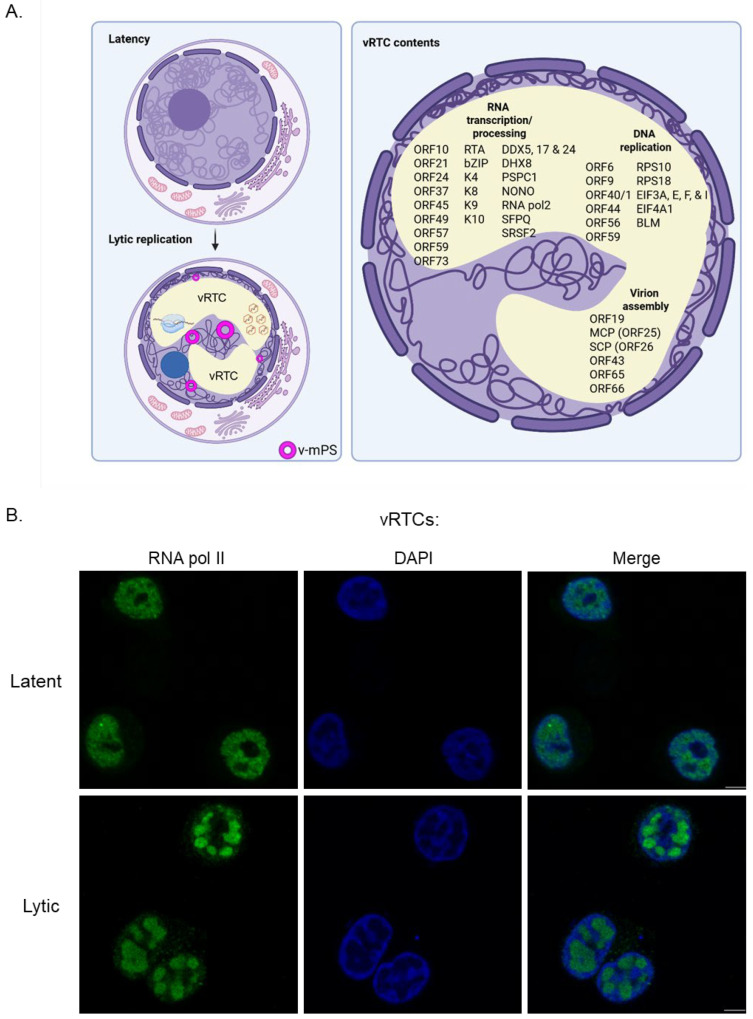
Dramatic effect on nuclear structure induced by KSHV lytic replication. (**A**) Schematic representation of the establishment of viral replication transcription centres (vRTCs) during lytic replication and how these dominate the nuclear space, compressing host chromatin to the periphery. Various sub-nuclear organelles such as nuclear speckles (NS) and virally modified paraspeckles (v-mPS) enlarge and closely associate with vRTCs during replication, while others, such as the nucleolus, are disrupted entirely. vRTCs contain both viral and host proteins to perform the three functions required: transcription of viral genes, DNA replication of the viral genome and assembly of virion cores. Proteins associated with or within vRTCs are annotated and split across these distinct processes and drawn from multiple mass-spectrometry-based analyses [8]. (**B**) Immunofluorescence of TREx-BCBL1-RTA cells during both latent and lytic (24 h post reactivation) lifecycles. vRTCs are highlighted with RNA pol II (green), and the compressed chromatin is highlighted by DAPI (blue), with methods performed as previously described, with KSHV reactivation was induced via the addition of doxycycline [16]. The schematic was created in BioRender. Whitehouse, A. (2025) https://BioRender.com/7yzgd1o (accessed on 1 October 2025).

### 2.2. How Does vRTC Formation Affect Sub-Nuclear Organelles?

There is emerging evidence that, alongside the dramatic compression of host chromatin, KSHV manipulates an array of other structures present within the nucleus. As such, KSHV provides a model of how membrane-less sub-nuclear organelles can be manipulated through the targeting of multiple structures during replication. These sub-nuclear organelles can be enlarged in size and redistributed as the vRTCs develop, presumably supporting or suppressing viral replication. Although their manipulation and potential roles, whether pro- or antiviral, remain to be fully understood. This therefore provides an intriguing model to understand the endogenous function of sub-nuclear organelles and how viruses can manipulate their function during infection.

#### 2.2.1. Nuclear Speckles and Paraspeckles

The terms nuclear speckles (NS) and paraspeckles refer to distinct sub-nuclear structures found within the nucleus of eukaryotic cells, with different compositions and biological roles. Nuclear speckles are rich in pre-mRNA splicing factors such as serine-/argine-rich proteins and serve as storage and modification sites for splicing factors, as well as regulating gene expression by modulating the availability of splicing machinery [17]. In contrast, paraspeckles are built around the architectural RNA (ArcRNA), *NEAT1*, and contain various core paraspeckles proteins such as NONO and SFPQ [18]. Their role is yet to be fully determined, but they are thought to regulate gene expression under stress, retaining certain RNAs and preventing their translation. Other roles in viral defence, regulating differentiation and miRNA processing have also been suggested [19].

Recent findings suggest that nearly one-third of KSHV genes express spliced transcripts, of which many undergo alternative splicing [20]. It would therefore appear necessary for KSHV to manipulate multiple sub-nuclear organelles to potentially support the production and modification of these transcripts. During lytic replication, NS have a dynamic relationship with vRTCs, increasing in size during viral replication, peaking to a maximum of 27 µm^3^ in volume compared to 4 µm^3^ in control untreated cells, while maintaining a peripheral location around the site of viral DNA replication (Figure 2A) [21]. During KSHV lytic replication, the components of NS are modified, splicing factors such as SRSF2 are maintained, whilst other components, such as the lncRNA MALAT1, are excluded, both of which are classical markers of canonical NS [21]. Recent evidence has suggested a role for the viral *kaposin* transcript in the remodelling and seeding of NS. Upon transcription, NS components are recruited to the viral transcript, suggesting that *kaposin* may act as an Architectural RNA (ArcRNA), seeding NS proximal to viral DNA, helping to optimise KSHV gene expression across all phases of infection [22]. This is a novel function of a viral RNA, which likely warrants further exploration to identify if other transcripts provide similar functions for other condensate structures during infection.

Induction of lytic replication results in a dramatic modification of paraspeckles; these sub-nuclear organelles relocalise to the periphery of vRTCs. Notably, the numbers of paraspeckles decrease but significantly increases in size from an average of ~700 nm to ~1800 nm, with the largest up to 3000 nm in diameter, which is 10× larger than canonical paraspeckles [16] (Figure 2B). These virus-modified structures retain condensate characteristics; however, virally modified PS (v-mPS) contain both viral proteins and transcripts, as well as an altered set of host factors compared to canonical paraspeckles [16]. Recent findings suggest that v-mPS may function as hubs for both viral RNA processing and host circRNA biogenesis, which in turn may regulate ncRNA regulatory networks during infection [16,23]. This is supported by depletion of core paraspeckle proteins, such as SFPQ and NONO. These proteins are required for v-mPS formation, and their depletion had a corresponding effect on inhibiting virus RNA processing and ultimately KSHV lytic replication and infectious virion production [16].

Therefore, manipulation of host cell splicing hubs appears to be an important requirement for KSHV replication, as many spliced transcripts from its genome were found to be localised to NS and v-mPS during infection [16,21]. In addition to the contribution to alternative splicing of viral genes, a secondary effect of manipulating these structures is a resulting increase in the genetic instability of host genes. KSHV lytic replication leads to an increase in R-loop formation and the relocalisation of SFPQ and other factors involved in DNA damage responses may be a factor in R-loop formation or a reduction in their resolution, indirectly playing a role in gamma herpesvirus-mediated oncogenesis [16,24].

#### 2.2.2. PML Nuclear Bodies

Promyelocytic leukaemia-nuclear bodies (PML-NBs) are a sub-nuclear organelle which restricts replication across a wide range of viral families [25]. The intrinsic and innate immune functions of PML-NBs and their core proteins are well defined for nuclear replicating viruses such as herpesviruses, papilloviruses, polyomaviruses, and adenoviruses [26], but they also play roles in the suppression of RNA virus replication, such as Zika, Dengue and Hepatitis C viruses [27]. PML-NBs function by multiple mechanisms, sequestering viral proteins, sumoylating proteins to alter stability and function, promoting apoptosis and autophagy and enhancing interferon signalling. Not surprisingly, both DNA and RNA viruses have evolved strategies to counteract these defences [28]. Intriguingly, PML-NBs appear to have the ability to inhibit herpesvirus replication across multiple phases of the viral lifecycle from initial infection to regulating gene expression and ultimately nuclear egress [29,30,31]. Concurrently, herpesviruses have evolved a multifaceted response to combat PML-NB restriction through the disruption and degradation of these structures. For example, KSHV protein vIRF3 has been shown to target PML itself for degradation through sumoylation [30]. Moreover, the KSHV ORF75 protein specifically leads to a redistribution of the canonical PML-NB component, ATRX, enhancing virus replication [32]. A novel antiviral role for PML-bodies recently observed during human cytomegalovirus (HCMV infection), using correlative light and transmission electron microscopy (CLEM), is the restriction of viral capsids by cage structures formed by PML-NBs [31]. While entrapment of viral capsids has been observed for other herpesviruses, Herpes Simplex Virus-1 (HSV-1) and Varicella Zoster Virus (VZV), whether or not this is a mechanical function enacted by the cellular defences against KSHV infection remains to be seen. However, this example serves to highlight the wide array of functions that a nuclear body can perform in an attempt to limit viral replication [31,33].

#### 2.2.3. Stress Granule Modulation

Modulation of condensate structures by KSHV, while predominantly nuclear-focused, also extends into the cytoplasm. Stress granules (SG) function as part of the host innate immune response to viral infection, acting as dynamic storage facilities for mRNA during times of cellular stress [34]. KSHV ORF57 and SOX proteins inhibit the formation of SGs during KSHV infection. ORF57 is a multifunctional protein, capable of nucleocytoplasmic shuttling and regulating multiple aspects of viral RNA processing. It also disrupts the interaction between protein kinase R (PKR) and PKR-activating protein, preventing the induction of SG formation through phosphorylation of elongation initiation factor (eIF) 2α [35,36]. Moreover, SOX-mediated endonuclease activity also inhibits the formation of SGs through the degradation of RNA in a blunt force manner [36]. This dual attack on the induction of SGs during replication highlights the redundancy often seen with KSHV-mediated dismantling of the cellular antiviral immune response. Interestingly, emerging evidence suggests the formation of PS is related to that of SGs, suggesting a link between the regulation of nuclear and cytosolic condensate structures [37]. The modulation of both SG and PS by KSHV may therefore be a more interrelated process than previously imagined and points to a relationship which likely requires further investigation to fully establish.

#### 2.2.4. Nucleolus

KSHV-mediated changes in nuclear architecture also lead to the disruption of the largest sub-nuclear organelle, the nucleolus [38]. Ordinarily, nucleolar structure is defined in three compartments, the fibrillar centre (FC), dense fibrillar component (DFC) and granular component (GC), with each having different concentrations of specific proteins and other components aiding in the sequential formation of ribosomes [39]. Importantly, the correct localisation of nucleolar factors is essential for healthy cell homeostasis, whereas nucleolar stress leads to the redistribution of factors, such as nucleoplasmic B23 [40]. Notably, maintained nucleolus stress leads to nucleolar shut-off, inhibited rRNA processing and eventually p53-mediated apoptosis [41].

During KSHV lytic replication, the nucleolus undergoes drastic morphological changes, with redistribution of many core nucleolar proteins to the nucleoplasm (Figure 3) [42]. Whilst this is a hallmark of nucleolar stress, nucleolus shutdown fails to occur, with little difference in rRNA processing and levels observed [42]. This is probably due to KSHV-mediated manipulation of ribosomal biogenesis, producing specialised ribosomes that preferentially translate KSHV encoded transcripts (An area further discussed below) [43,44]. Therefore, it is likely beneficial to KSHV to maintain nucleoli functionality, although how it circumvents the stress pathways is unknown. Notably, several KSHV-encoded proteins can localise to the nucleolus, encoded by ORF57, ORF11 and ORF20, which may play a role in regulating nucleolar stress [42,45,46].

## 3. What Impact Does KSHV Lytic Replication Have on Nuclear-Related Pathways?

### 3.1. Ribosomal Biogenesis and Translation Control

Ribosomes are large macromolecular machines, comprising rRNA and over 80 complexed ribosomal proteins, tasked with the translation of mRNA transcripts into polypeptide chains for downstream folding and production of mature proteins. Like all viruses, KSHV lacks its own translational machinery and therefore co-opts cellular ribosomes for viral protein production. During latency, KSHV is predominantly translationally dormant, only expressing a few latency-associated proteins; however, upon induction of the lytic replication phase, rapid viral ORF translation places a high demand on host cell ribosomes. Quantitative proteomic analysis during KSHV lytic replication showed a broad reduction in host protein levels [48], which may be linked to host shut-off-related mechanisms, although this has not been specifically shown. Host cell shutoff-related mechanisms are thought to allow the virus to redirect cellular machinery to prioritise viral gene expression and protein synthesis. For example, the KSHV-encoded host cell shut-off protein, SOX, degrades ~80% of host mRNA transcripts [49] and the KSHV ORF57 protein sequesters cellular RNA processing factors to viral mRNAs, enhancing their stability and nuclear export [50].

KSHV also modifies aspects of the translational pathway to selectively upregulate viral protein production. Translation is split into four main stages: initiation, elongation, termination and recycling. Like many viruses, current research indicates that KSHV predominantly targets translation initiation to regulate cellular and viral protein output to favour virus replication through multiple mechanisms (summarised in Figure 4). The translation initiation factor eIF4E binds to the 5’ cap structure of mRNA transcripts as part of the eIF4F complex, which recruits the 40S ribosomal subunit for translation initiation. 4E-BP1 is a translational repressor sequestering eIF4E, preventing eIF4F complex formation [51]. KSHV lytic replication mediates the inactivation of 4E-BP1, freeing up eIF4E to increase translational output (Figure 4) [52]. The immediate-early viral protein encoded by ORF45 alters several cellular processes, including translation initiation, through the activation of signalling pathways. Specifically, ORF45 activates the ERK/RSK pathway by interacting directly with p90 ribosomal s6 kinase (RSK) and stimulating its kinetic activity [53]. Phosphorylation of eIF4B via the ORF45/RSK axis leads to enhanced formation of the pre-initiation complex and increased translation [54]. Moreover, RSK1 is sumoylated by ORF45 at Lys^110^, Lys^335^ and Lys^421^, required for the recruitment and subsequent phosphorylation of eIF4B, leading to enhanced translation initiation and thus an increased translation of viral mRNA (Figure 4) [54,55]. The nucleocytoplasmic shuttling protein ORF57 interacts with both the 40S ribosomal subunit and the translation enhancement factor PYM. ORF57 directly associates with viral intronless mRNAs mediating the recruitment of the preinitiation complex via PYM, to enhance translation [46]. Moreover, ORF57 co-sediments with actively translating polysomes enriched with poly(A)binding protein 1 (PAPB1) and a decreased association of Ago2, a core component of the RNA-induced silencing complex (RISC). It is likely that Ago2 reduces translational output by degrading the mRNA associated with polysomes; therefore, by preventing this association, ORF57 is enhancing protein translation and stability (Figure 4) [56]. Additionally, an intriguing mechanism of translational control has been linked to KSHV-upregulation and relocalisation of HIF2α to the endoplasmic reticulum, which enables access to the alternative translation machinery eIF4F^H^. This so-called “translation initiation plasticity” allows translation of viral and host proteins via mTOR-dependent or -independent mechanisms during KSHV lytic replication [57].

Alongside enhancing traditional ribosomal functioning, KSHV can generate modified ribosomes, supporting the specialised ribosome hypothesis [43]. Increasing reports have identified that modifications to ribosomal proteins, rRNA, or the structure and stoichiometry of ribosome-associated proteins can regulate translation of specific transcripts [58]. The previously uncharacterised KSHV ORF11 protein associates with cellular ribosomes during lytic replication and mediates the increased association of several ribosomal biogenesis factors with the pre-40S subunit during the early stages of biogenesis, namely the BUD23/TRMT112 and NOC4L/NOP14/EMG1 complexes [43]. Interestingly, both these complexes contain methyltransferases, which increase methylation of specific residues on the 18S rRNA during KSHV lytic replication. BUD23 mediates the N7-methylation of G1639, whereas EMG1 catalyses the N1-methylation of 1248Ψ [59,60]. Depletion of BUD23 leads to an increase in viral uORF translation over the main coding sequence (CDS) and has led to the hypothesis that m7G1639 enhances scanning of the pre-initiation complex along mRNA, thus preventing ‘leaky’ scanning of non-cognate start codons (Figure 4) [43]. EMG1 association with pre-40S ribosomal subunits also increases during KSHV lytic replication in a phenotype similar to that observed for BUD23 [44]. The methylation activity of EMG1 was demonstrated to be required for the proper translation of KSHV CDSs, with the loss of EMG1 resulting in decreased translation of viral genes and concomitant reduction in viral replication [44].

### 3.2. RNA Modification Pathways

The epitranscriptome is defined by a broad family of over 100 different chemical modifications on RNA molecules that have critical roles in regulating the fate of every RNA species [61]. As viruses have evolved to maximise their coding capacity and to utilise host–cell machinery, they also target many of these RNA modifications as another layer of host–cell virus manipulation.

N^6^-methyladenosine (m^6^A) is a highly abundant RNA modification in eukaryotic RNAs found in almost all RNA species, including circular RNAs (circRNAs), long non-coding RNAs (lncRNAs), ribosomal RNAs (18S and 28S RNA), microRNAs (miRNAs), small nuclear RNAs (snRNAs) and is the most prevalent internal modification of eukaryotic mRNAs. The modification is thought to be dynamic, characterised by the addition, reading or removal of m^6^A by the so-called writers, readers or erasers, respectively [62] (Figure 5). The addition of m^6^A methylation occurs co-transcriptionally in the nucleus and is canonically catalysed by the METTL3-METTL14 methyltransferase complex, methylating the central adenosine residue at the consensus DRACH sequence (D=A/G/U, R=A/G and H=A/C/U) [63,64,65]. Additional components, such as WTAP, RBM15, VIRMA and ZC3H13, are responsible for selectivity, localisation and structural integrity of the writer complex [64,66,67]. These interactions localise METTL3-METTL14 complex into nuclear speckles, allowing co-transcriptional deposition of m^6^A on DRACH motifs [68]. When deposited, the m^6^A modification is then recognised by reader proteins, primarily of the YTH domain-containing family. The YTH domain contains a conserved aromatic cage that recognises and binds to m^6^A in a sequence-independent manner [69]. In contrast, a second group of m6A readers, including several hnRNPs, preferentially bind m6A-modified RNAs through an m6A switch mechanism, an event in which m6A modification remodels local RNA structure [70]. A myriad of m6A readers may therefore exist, which enable widespread regulatory control over gene expression, affecting many biological pathways [71]. Two RNA demethylases act as m6A erasers, α-ketoglutarate-dependent dioxygenase alkB homologue 5 (ALKBH5) and fat mass obesity protein (FTO) revert m^6^A back to adenosine residues [71,72,73]. Although the impact and extent of demethylation carried out by the m6A ‘erasers’ ALKBH5 and FTO is debated, it is widely accepted that at least some m6A residues can be reversed back to adenosine [74].

KSHV has been shown to directly influence the m^6^A landscape within the infected cell. During lytic reactivation, there is a marked global decrease in m^6^A deposition on cellular RNAs and a significant increase in the deposition of m^6^A on KSHV RNA species [75,76]. This global decrease is hypothesised to be potentially driven by SOX-mediated RNA decay of cellular transcripts; therefore, fewer transcripts are present overall. Alternatively, this global change could be driven by specific manipulation of m^6^A machinery to influence preferential m^6^A deposition on KSHV transcripts. These changes could be due to alterations in m^6^A machinery expression, localisation or changes to the machinery’s interactome. This marked reduction in m^6^A deposition on cellular transcripts could influence the stability or translation ability of mRNAs encoded by genes involved in innate immunity, thus generating a more favourable replication environment for the virus. In contrast, a subset of cellular mRNAs has been shown to be heavily increased in both m^6^A content and abundance during KSHV lytic replication, such as GPCR5, to enhance virus replication [77].

The influence of m^6^A on KSHV RNA transcripts is determined by the binding of specific reader proteins. The m^6^A reader YTHDC1, in complex with SRF3 and SRS10, recognises a single m^6^A site on the pre-mRNA of KSHV latent-lytic switch protein Replication and transcription activator (RTA) transcript [76]. This site was shown to be essential for its splicing into a mature functional mRNA. Moreover, SND1, a protein from the class of m^6^A readers known as the “Royal” family (named after the presence of a “Tudor” m^6^A binding domain), was shown to affect the stability of unspliced RTA RNA [78].

Another key m^6^A reader, YTHDF2, has both pro-viral and anti-viral roles within KSHV replication, depending on cell type [79]. Upon YTHDF2 depletion in TREx-BCBL1 cells, increased expression of RTA was observed. In a contrasting study, however, YTHDF2 depletion in iSLKs resulted in decreased RTA expression and thus a decrease in overall viral replication [79]. The discrepancy between these two findings highlights the dynamic nature of m^6^A, indicating that KSHV-mediated manipulation of m^6^A pathways may shift depending on cell type.

Interestingly, it has also been shown that m^6^A plays an intrinsic role in the protection of Interleukin-6 (IL-6) against SOX-mediated decay [80]. IL-6 gains its SOX resistance through the binding of YTHDC2 on m^6^A-modified IL-6 transcripts. Furthermore, the deposition of m^6^A and subsequent resistance to SOX-mediated decay only occur during KSHV lytic infection, defining an essential mechanism of how KSHV manipulates the m^6^A pathway to regulate both viral and cellular gene expression [80]. Despite this, m^6^A is not the sole contributor to SOX resistance, as many SOX-resistant mRNAs had no change in their m^6^A status during lytic reactivation. Therefore, it is hypothesised that the specific deposition of m^6^A must be contextual to recruit a specific set of protective reader proteins to be active [80].

KSHV not only manipulates the m^6^A pathway but also a range of other epitranscriptome-related factors. Adenosine-to-Inosine (A-to-I) editing utilises adenosine deaminases (ADARs) to hydrolytically deaminate adenosine, resulting in inosine [81]. Inosine preferentially base-pairs with cytosine, allowing for significant effects on gene expression. This modification can affect pre-mRNA splicing, transcript stability and recoding of protein sequences. A-to-I editing can affect viral replication through the suppression of the IFN response to endogenous RNA [81,82]. Alternatively, A-to-I editing can have anti-viral or pro-viral effects depending on alterations in certain RNA expression dynamics [83]. During reactivation, KSHV increases adenosine deaminase activity, resulting in not only a large increase in the number of sites edited but also the editing frequency of previously edited sites. This increase in activity is not due to increased expression of ADAR1/2 but hypothesised to be a result of ADAR1 relocalisation to the cytoplasm and KSHV activation of p38-MSK1&2 MAP kinases [84]. Furthermore, ADAR1 was shown to be responsible for preventing innate immune activation during reactivation and depletion of ADAR1 resulted in enhanced IFNB production [85]. While the mechanisms and networks regulating A-to-I editing are understudied in KSHV, it is clear that the virus can manipulate specific editing for pro-viral functions.

Recent research has revealed the presence of the *N4*–acetylcytidine (ac^4^C) modification in the KSHV transcriptome [86]. ac^4^C consists of acetylation at cytidine N^4^ position on RNAs that can be catalysed by N-acetyltransferase 10 (NAT10). Modification by NAT10 is essential for the upregulation of KSHV viral lytic gene expression and virion production, through stabilisation of KSHV polyadenylated nuclear RNA (PAN). Interestingly, through interactions with NAT10, PAN was shown to assist in the ac^4^C modification of the nuclear pathogen sensor IFI16 RNA, resulting in increased stability and expression of IFI16 and activation of the inflammasome. Interestingly, the modification landscape of PAN has been further elucidated through the identification of pseudouridine (Ψ), an RNA modification characterised as a C5-glycoside isomer of uridine. Ψ is proposed to confer similar phenotypes to PAN as ac^4^C, affecting KSHV viral replication and PAN transcript expression [87]. This highlights the extraordinary depth of the RNA modification landscape in DNA viruses.

### 3.3. Epigenetic Control Mechanisms

The KSHV episome is heavily interlinked with the epigenetic control mechanisms of the host cell through the deposition and removal of histone marks, chromatin remodelling and post-transcriptional processing in a complex balance of gene regulation (Reviewed extensively in [88,89]. During latency, transcription of most viral genes is suppressed, partly thanks to the deposition of suppressive histone marks, such as H3K27me3, at promoters and transcriptional start sites as well as spatial regulation by chromatin looping [90,91]. Suppression of the vast majority of viral transcripts avoids immune detection and the resulting clearance of latently infected cells [92]. There is, however, a small subset of viral genes which remain actively transcribed during latency to maintain the viral episome and promote latently infected cell survival. LANA, a multifunctional protein responsible for tethering viral episomes to host chromatin, is one such transcript expressed during latency and displays enrichment of pro-transcriptional histone markers such as H3K4me3 [90,93]. Upon viral reactivation, a lytic cascade of gene expression ensues, beginning with transcripts which do not require de novo expression of viral genes prior to their transcription. These immediate early genes are marked with bivalent chromatin modifications, which remain suppressed during latency and yet are primed for transcription upon triggering of lytic replication [93]. Subsequent early and late gene expression is then under the control of the preceding viral genes to continue the cascade. Disruption of epigenetic control can trigger lytic replication through the removal or inhibition of methylation of suppressive histone modifications on the viral episome [94]. This mechanism of inducing lytic replication is used across many KSHV cell models, where the addition of chemicals such as sodium butyrate and tetradecanoyl-phorbol-13-acetate results in the inhibition of suppressive histone marks and promotes chromatin remodelling in the immediate early genes, including RTA, the protein driver of lytic reactivation [95].

### 3.4. DNA Damage-Related Pathways

One of the key cellular mechanisms occurring within the nucleus is the DNA damage response (DDR) mediated through multiple different pathways, including but not limited to non-homologous end joining (NHEJ) and homologous recombination (HR), both of which are involved with repairing double-stranded breaks (DSBs), with NHEJ being the primary pathway [96,97]. Throughout its lytic replicative cycle, KSHV interacts and dysregulates multiple aspects of the DNA damage response pathways, with these interactions between the virus and the host aiding in viral maintenance and replication while also contributing to KSHV-mediated oncogenesis.

During KSHV lytic infection, there is a well-documented increase in DSBs, with increases in classical markers such as phosphorylated ATM (a DDR sensor protein), DNA-PKcs and phosphorylated H2AX (Figure 6) [98,99]. This induction of DNA damage is not just an indirect consequence of KSHV replication, but instead a potential pro-viral strategy. This is highlighted by the inhibition of ATM, which leads to a reduction in KSHV replication [98], whereas inhibition of DNA-PK results in increased ATM phosphorylation and a corresponding increase in KSHV virion production, with a similar phenotype noted in EBV-infected cells [98,100]. Not only is phosphorylation of ATM itself important for replication, its downstream interactors are also targeted by KSHV. For example, the viral protein vIRF1 can directly bind to ATM, preventing the phosphorylation of p53 at Ser51, leading to a reduction in p53 protein levels and subsequent effects therein [101]. The DNA processivity factor, ORF59, binds to the NHEJ proteins Ku70 and Ku80, reducing the recruitment of DNA-PKcs, leading to an increase in DSBs due to a reduction in the speed and efficiency of the DDR (Figure 6) [99].

KSHV also redistributes host-encoded DNA damage proteins, such as RPA32 and MRE1, to sites of viral replication located in the viral replication centres, and while their exact role in KSHV replication remains unclear, these DDR factors are essential for DNA replication in other herpesviruses (Figure 6) [98,102,103]. KSHV has also been shown to sequester host proteins within novel structures, such a v-mPS, during lytic replication [16]. During this process, the splicing factor proline and glutamine-rich (SFPQ) is relocalised and concentrated away from its canonical environment [16]. As a result, SFPQ has a reduced interaction with Ku70, leading to a reduction in NHEJ efficiency and an accumulation of dsDNA breaks [16,104,105,106].

DSBs can also be induced through the aberrant hybridisation of newly transcribed mRNAs with the genomic template, forming DNA–RNA hybrids, known as R-loops [107]. A dramatic increase in R-loop formation is observed during KSHV replication, leading to an increase in DNA damage [24]. The viral protein, ORF57, is suggested to play a contributing role in this increase in R-loops through its recruitment of the human transcription and export complex (hTREX) onto viral RNAs, sequestering hTREX away from cellular mRNAs [24,108]. This series of interactions and recruitment of hTREX facilitates the nuclear export of viral intronless transcripts, which would otherwise fail to be trafficked efficiently to the cytoplasm for translation [46,108]. However, a consequence of hTREX complex sequestration to viral mRNAs is a reduction in the availability of the complex to prevent RNA/DNA hybrids from being formed from newly transcribed cellular mRNA, increasing R-loop formation (Figure 6) [24].

### 3.5. Tumourigenesis-Related Pathways

With such drastic nuclear remodelling, disruption to DNA damage repair functions and manipulation of the epitranscriptome, it is unsurprising that oncogenic outcomes occur with KSHV infection. Oncogenesis is further increased by the expression of multiple virally encoded oncoproteins during both the latent and lytic replication phases [1]. Although latency-associated KSHV proteins have been well characterised in persistence, transformation and tumourigenesis pathways [109], it is now evident that the KSHV lytic cycle also contributes to oncogenesis, which could provide important targets in the development of anti-cancer therapeutics [1]. This is supported by the treatment of KS patients with drugs that prevent lytic replication and, in certain cases, lead to regression of KS lesions; attesting to the importance of lytic gene expression in tumourigenesis [110,111]. However, how latent persistence and the lytic phase, which ultimately leads to host cell lysis, co-exist and interact to induce the transformation programme is yet to be fully elucidated.

Although only a small percentage of cells within the KS tumour undergo lytic replication, they are thought to have multiple roles in tumourigenesis. This includes enhancing the spread of the virus from latent B cells to endothelial cells, where tumours arise [112]; sustaining the latently infected tumour cell population that would otherwise be reduced due to the poor persistence of the KSHV episome during tumour cell division [113]; and lytic KSHV-induced paracrine secretion of pro-inflammatory and angiogenic factors essential for tumour development, modulating the behaviour of latently infected cells, and potentially the tumour microenvironment [114,115].

Lytic oncoproteins, including vGPCR and K1, induce pro-proliferation signalling pathways [115]. vGPCR induces many of the same pathways as its cellular homologue, including ERK and PI3K/AKT/mTOR [116,117,118]. The induction of this myriad of signalling pathways activates Sp1/3 transcription factors, which are crucial for the maintenance of RTA expression, committing the cell to productive lytic replication [119,120]. Whilst K1 activates pro-proliferation signalling pathway activity, including PI3K/Akt activity [121,122]. Viral interleukin 6 (vIL-6), a homologue of human IL-6, is another lytically expressed gene which plays a multifaceted role in tumourigenesis (extensively reviewed in [123]). vIL-6 is highly expressed in KSHV lytically replicating cells and is also expressed in a small population of latently infected cells at low concentrations. It plays a significant role in tumorigenesis by promoting cell proliferation, angiogenesis, and cell migration, as well as inhibiting the immune response. Mechanistically, it activates key signalling pathways similar to those of human cytokines, namely JAK/STAT, Ras/MAPK, PI3K/Akt, leading to the suppression of tumour suppressors such as caveolin-1. Additionally, vIL-6 helps regulate viral gene expression and contributes to a pro-inflammatory environment that supports tumour development. The KSHV genome also possesses three lytically encoded viral interferon regulatory factors (vIRFs), which disrupt the antiviral IFN response by inhibiting transcription of IFN and inflammatory signals, in turn leading to the dysregulation of the IFN antiviral response, apoptosis and cell cycle arrest, ultimately increasing the oncogenic potential of KSHV-infected cells [124,125].

An alternative role of the lytic phase in tumourigenesis has also been postulated, where a dysregulated viral transcription programme occurs, distinct from either latent or lytic phases, termed the abortive lytic phase [118]. Here, sporadic expression of lytic genes is observed without infectious virion production and cell lysis [126], driven by epigenetic regulation of the KSHV and host epigenome, leading to upregulation of viral oncogenes and downregulation of tumour suppression and innate immune genes, which would curtail growth [126,127]. Therefore, the overall combination of both latent and lytic lifecycles drives KSHV-mediated tumorigenesis, and understanding host cell manipulation throughout these cycles will aid in novel therapeutics.

## 4. Summary

KSHV manipulation of the nuclear space during lytic replication fosters an environment favourable for production of viral transcripts, genomes and progeny. Establishing multifunctional vRTC structures results in dramatic morphological changes in the nuclear architecture and provides a hub for viral lifecycle progression. Manipulation of multiple sub-nuclear organelles by the virus enables the modulation of a multitude of cellular processes, from innate immune sensing, for example, disrupting PML-NBs, to the modification of paraspeckles, to supporting complex RNA processing events required for viral replication. KSHV also manipulates multiple nuclear-related pathways, such as the epitranscriptome and translational initiation machinery, modifying transcript methylation, base incorporation and specialised ribosomal biogenesis to facilitate efficient replication. The DNA damage machinery, induction and response are also affected by KSHV due to the manipulation of host proteins and disruption of their canonical interaction partners and processes, which can facilitate tumorigenesis. Therefore, the lack of effective treatments or anti-viral regimes requires the continual study and understanding of the processes KSHV uses to manipulate the host cell environment to identify new therapeutic strategies.

## Figures and Tables

**Figure 2 viruses-17-01427-f002:**
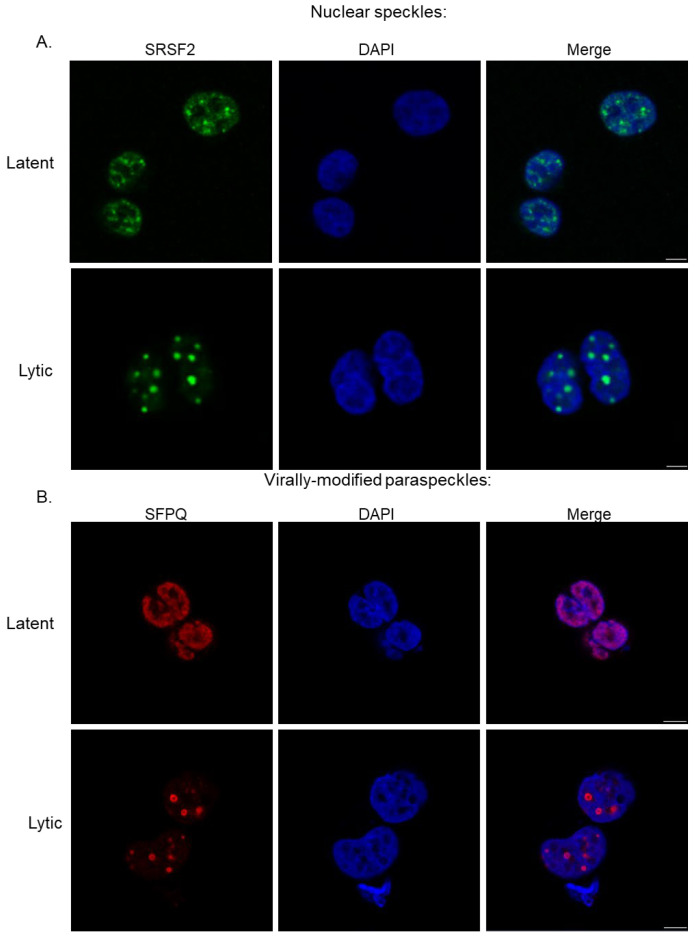
Immunofluorescence of latent and lytic TREx-BCBL1-RTA cells highlighting (**A**) nuclear speckles (SRSF2, green) and (**B**) virally modified paraspeckles (SFPQ, red). DAPI is stained blue. Cells were reactivated via doxycycline as previously described [16]. Similar results have also been observed in iSLK cells during KSHV lytic replication [22].

**Figure 3 viruses-17-01427-f003:**
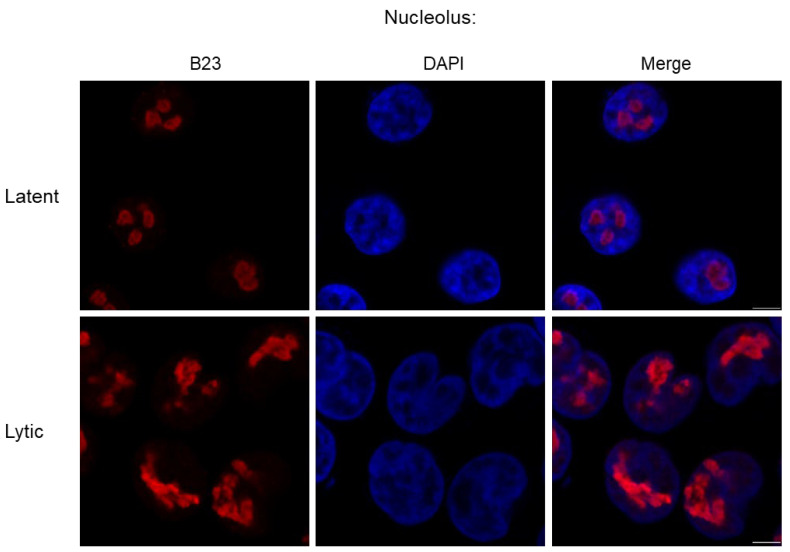
Immunofluorescence of latent and lytic TREx-BCBL1-RTA cells, highlighting the nucleolus utilising B23 (red) and DAPI (blue), with IF performed as previously described, cells were reactivated with doxycycline [47]. Similar dysregulation is also observed in iSLK cells during KSHV lytic replication [42].

**Figure 4 viruses-17-01427-f004:**
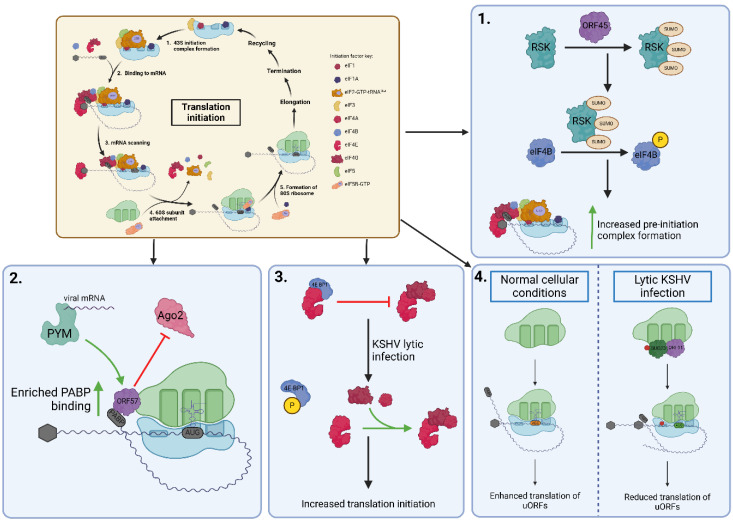
KSHV manipulation of translation initiation. Translation initiation is a highly regulated mechanism in which translation initiation factors (eIFs) bind to the 40S ribosomal subunit to form the pre-initiation complex. This scans along mRNA until a start codon is recognised and the 60S ribosomal subunit is recruited, thus triggering dissociation of eIFs and assembly of the 80S ribosome for the next stage of translation. During lytic KSHV infection, the virus co-opts several aspects of translation initiation to promote translation of viral transcripts. (**1**) Viral protein ORF45 induces SUMOylation of p90 ribosomal s6 kinase (RSK), resulting in the direct recruitment and phosphorylation of the initiation factor eIF4B. This enhances the formation of pre-initiation complexes and leads to increased translation initiation. (**2**) The viral protein ORF57 has enhanced binding with polyA binding protein (PABP) to enhance translational output whilst preventing Ago2 binding to polysomes and causing mRNA degradation. Moreover, ORF57 also binds to PYM, which is able to recruit spliced KSHV mRNA transcripts to ribosomes for translation. (**3**) KSHV lytic replication causes phosphorylation of the translational repressor 4E-BP1, which in turn frees the initiation factor eIF4E to bind to the other components of the eIF4F complex and increase translation initiation. (**4**) The viral protein ORF11 mediates the formation of specialised ribosomes by recruiting methyltransferase BUD23 to the pre-40S subunit. Subsequent methylation enhances accuracy in pre-ribosomal subunit scanning, preventing the recognition of non-cognate start codons and translation of uORFs. Created in BioRender. Whitehouse, A. (2025) https://BioRender.com/33iefg8 (accessed on 1 October 2025).

**Figure 5 viruses-17-01427-f005:**
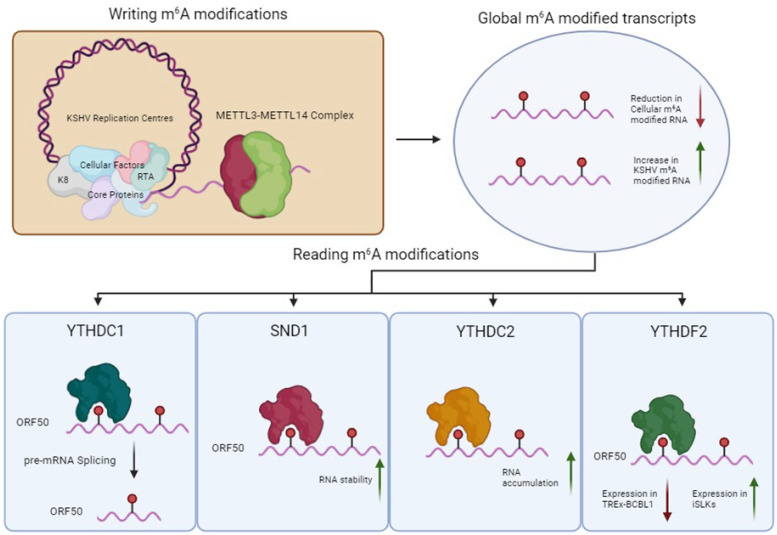
KSHV manipulation of m^6^A modification pathways. KSHV preferentially modifies KSHV transcripts, resulting in a global shift of m^6^A from cellular to KSHV transcripts. Readers such as YTHDC1, SND1, YTHDC2 and YTHDF2 are co-opted by KSHV to influence essential post-transcriptional functions of mRNA, such as pre-mRNA splicing, RNA stability, RNA accumulation and nuclear export. Created in BioRender. Whitehouse, A. (2025) https://BioRender.com/raafwte (accessed on 1 October 2025).

**Figure 6 viruses-17-01427-f006:**
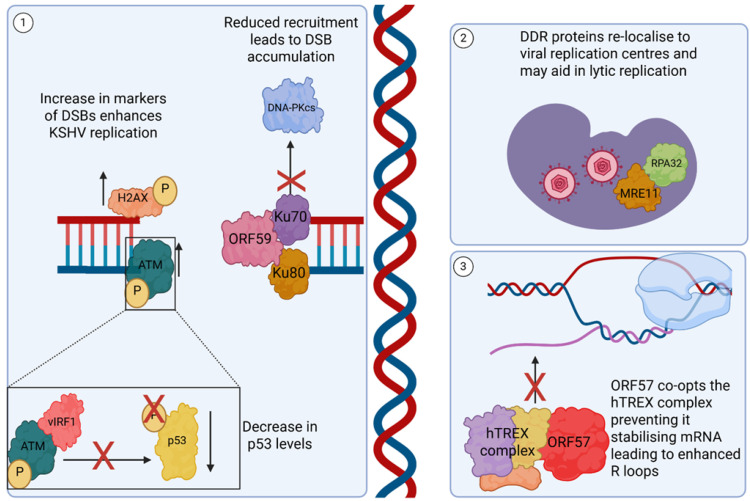
KSHV dysregulation of DNA repair machinery. (**1**) During KSHV infection, markers of DSBs increase in a pro-viral response, including phosphorylated H2AX and ATM. The viral protein vIRF1 binds to phosphorylated ATM and blocks its phosphorylation of p53 serine 51, leading to reduced levels of p53. Whilst another viral protein ORF59 binds to the Ku70/80 heterodimer, reducing its recruitment of DNA-PKcs, contributing to an accumulation of DSBs during lytic replication. (**2**) Other proteins involved in the DDR, including RPA32 and MRE11, are recruited during lytic replication to virus replication and transcription compartments (vRTCs). (**3**) ORF57 hijacks the hTREX complex to aid in the export of intronless KSHV transcripts. This subversion prevents individual hTREX components from stabilising mRNA during transcription, increasing the prevalence of DNA–RNA hybrids, termed R loops. Created in BioRender. Whitehouse, A. (2025) https://BioRender.com/randqnr (accessed on 1 October 2025).

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
