# Peer review of "Manipulation of Nuclear-Related Pathways During Kaposi’s Sarcoma-Associated Herpesvirus Lytic Replication"

_viruses, 2025, doi:10.3390/v17111427_

Round 1

Reviewer 1 Report

Comments and Suggestions for Authors

This is a comprehensive review which discuss the impact of KSHV lytic replication on the host cell nucleus and nuclear-related pathways. They focus on viral replication and transcription centers (vRTCs), DNA damage response, ribosomal biogenesis and epigenetic regulation which are disrupted or modified during KSHV replication. I have a few questions which hope can be addressed:

1) Please label Figure 1A and 1B in the panel, Figure 3A should be only Figure 3.

2) In Section 2.2, I hope they can explain what are the differences between NS and PS (v-mPS). e.g., Different markers and components? Different size? Do they appear in the nuclear at different time during lytic replication?

3) There are some experimental data especially IFA results shown in the figures. Are these data published or unpublished, if published please add the reference, if unpublished please provide the resource.

4) There are a few typo or format errors in the text which should be corrected.

Author Response

1) Please label Figure 1A and 1B in the panel, Figure 3A should be only Figure 3.

Corrected

2) In Section 2.2, I hope they can explain what are the differences between NS and PS (v-mPS). e.g., Different markers and components? Different size? Do they appear in the nuclear at different time during lytic replication?

This section has been modified, we have added an introductory paragraph explaining the differences in nuclear speckles and paraspeckles, including components, sizes, function etc, as requested.

3) There are some experimental data especially IFA results shown in the figures. Are these data published or unpublished, if published please add the reference, if unpublished please provide the resource.

The IFA images have not been published previously, we have included them to illustrate architectural changes. Similar findings have been published previously and we have now included these references in the figure legends. We have updated the figure legends to include further details of the experiments as requested.

4) There are a few typo or format errors in the text which should be corrected

Corrected throughout.

Reviewer 2 Report

Comments and Suggestions for Authors

The authors here provide a wide ranging and thorough review of what has become a topic of great interest to the field. I have provided a number of minor comments below for clarity or additional discussion. My two major comments are 1) that the review could be improved by further emphasis on driving or open questions for each section. And 2) that the authors should consider restructuring to follow the KSHV life cycle and focus on how these nuclear changes are thought to support or antagonize each of these steps (i.e. latency, early transcription, DNA replication, packaging etc) or where this unknown.

Minor comments:

Abstract “predisposes infected cells to oncogenic transformation” but the reviewed events primarily occur in in lytic cells?

Page 2: Should cite or discuss previous paper as showing ‘psuedo-RCs’. https://pubmed.ncbi.nlm.nih.gov/11152521/

Page 2: “it could therefore be predicted that they will contain some form of compartmentalization.” Not clear what the authors mean here.

Figure 1b: please label RNAPII and DAPI in the figure.

Page 4: Open question: how are these other compartments effected in overexpression of viral components sufficient to cause RC like structures.

Page 4: “Alternatively spliced transcripts make up a large proportion of the KSHV transcriptome during lytic replication.” This claims requires a reference or clarification. As typical KSHV transcripts are not spliced (though many (most?) KSHV genes do have minor spliceforms).

Page 4: “KSHV replication modifies NS.” Not clear what is referred to here. Is replication modifying the structure, compeonents, or contents of the NSs?

Page 4: “arcRNA” acronym needs to be spelled out

Page 4: If paraspeckles are a distinct organelle they would have their own section, or retitle it to splicing machinery or something similar.

Page 5: “may be an increase in genetic instability.” This requires additional clarification. For example: is this genetic instability of the virus or the host?

Page 5: Further discussion of the known antiviral mechanisms of PML bodies in other non-herpes viruses and how they might be related to the existing evidence in KSHV would be appropriate.

Page 7: “global host cell translation is dramatically reduced due to SOX-mediated host cell shut off…” I am not sure this statement is true (though it might be expected). The best evidence might be this https://pmc.ncbi.nlm.nih.gov/articles/PMC7567700/ where they do show a broad reduction in host protein levels. Though they do not show SOX dependency.

Page 7: The level of molecular detail for the translational initiation is noticeably higher than other sections. A less detailed summary might be helpful to guide the reader better in what is understood and what is still uncertain.

Translation manipulation should likely have its own section instead of being under sub-nuclear compartments.

Section 4 is probably unnecessary as it is thin here and has been reviewed elsewhere.

ALT is mentioned in the figure legend of Figure 6, but I don’t see any other explanation or mention of it in the text.

Page 15: “However, latency is insufficient for complete oncogenic transformation, with lytic replication also required” and “A full lytic replicative cycle leads to host cell death, therefore lytic replication probably contributes to oncogenesis through abortive lytic replication” are both controversial statements, and would require further explanation and support that is likely outside the focus of this review. Indeed section 6’s inclusion in the review perhaps should be reframed as nuclear manipulation during latency.

Author Response

Reviewer 2

My two major comments are 1) that the review could be improved by further emphasis on driving or open questions for each section.

We have added open questions to each main section and reduced the main sections with sections separated into sub nuclear organelles and nuclear-related pathways.

2) that the authors should consider restructuring to follow the KSHV life cycle and focus on how these nuclear changes are thought to support or antagonize each of these steps (i.e. latency, early transcription, DNA replication, packaging etc) or where this unknown.

We wish to retain the original structure, albeit modified as above, as each section encompasses different sections of the lytic phase of replication and think it is easier to focus on each subnuclear organelle and pathway separately.

Minor comments:

Abstract “predisposes infected cells to oncogenic transformation” but the reviewed events primarily occur in in lytic cells?

We have removed this sentence for clarity and focus.

Page 2: Should cite or discuss previous paper as showing ‘psuedo-RCs’. https://pubmed.ncbi.nlm.nih.gov/11152521/

Added and discussed.

Page 2: “it could therefore be predicted that they will contain some form of compartmentalization.” Not clear what the authors mean here.

Sentence added for clarification.

Figure 1b: please label RNAPII and DAPI in the figure.

Figure labels have been corrected on Figure 1 and 3.

Page 4: Open question: how are these other compartments effected in overexpression of viral components sufficient to cause RC like structures.

This is unknown.

Page 4: “Alternatively spliced transcripts make up a large proportion of the KSHV transcriptome during lytic replication.” This claims requires a reference or clarification. As typical KSHV transcripts are not spliced (though many (most?) KSHV genes do have minor spliceforms).

Corrected and reference added (Majerciak et al 2022 PLoS Pathogens).

Page 4: “KSHV replication modifies NS.” Not clear what is referred to here. Is replication modifying the structure, compeonents, or contents of the NSs?

Modified sentence to clarify statement.

Page 4: “arcRNA” acronym needs to be spelled out

Corrected.

Page 4: If paraspeckles are a distinct organelle they would have their own section, or retitle it to splicing machinery or something similar.

Tiltle of section modified and paragraph added to discuss differences between speckles and paraspeckles.

Page 5: “may be an increase in genetic instability.” This requires additional clarification. For example: is this genetic instability of the virus or the host?

Corrected and expanded upon.

Page 5: Further discussion of the known antiviral mechanisms of PML bodies in other non-herpes viruses and how they might be related to the existing evidence in KSHV would be appropriate.

Extra details included at start of PML-NB section.

Page 7: “global host cell translation is dramatically reduced due to SOX-mediated host cell shut off…” I am not sure this statement is true (though it might be expected). The best evidence might be this https://pmc.ncbi.nlm.nih.gov/articles/PMC7567700/ where they do show a broad reduction in host protein levels. Though they do not show SOX dependency.

This section has been modified by toning down the SOX link and adding the suggested reference.

Page 7: The level of molecular detail for the translational initiation is noticeably higher than other sections. A less detailed summary might be helpful to guide the reader better in what is understood and what is still uncertain.

This section has been reduced a little.

Translation manipulation should likely have its own section instead of being under sub-nuclear compartments.

We have now separated the sections into subnuclear organelles and nuclear-related pathways. Ribosomal biogenesis and translation control is now under nuclear-related pathways.

Section 4 is probably unnecessary as it is thin here and has been reviewed elsewhere.

We would like to retain this section, but now include it as a section under nuclear-related pathways and not an independent section.

ALT is mentioned in the figure legend of Figure 6, but I don’t see any other explanation or mention of it in the text.

We have removed ALT from Figure 6, to align with the text.

Page 15: “However, latency is insufficient for complete oncogenic transformation, with lytic replication also required” and “A full lytic replicative cycle leads to host cell death, therefore lytic replication probably contributes to oncogenesis through abortive lytic replication” are both controversial statements, and would require further explanation and support that is likely outside the focus of this review. Indeed section 6’s inclusion in the review perhaps should be reframed as nuclear manipulation during latency.

This section has been extensively modified to expand on the role of the KSHV lytic replication cycle in more detail. We wish to retain the focus on lytic replication in line with the rest of the review.

Reviewer 3 Report

Comments and Suggestions for Authors

In this review, Connor Hayward et al. described the Manipulation of nuclear-related pathways during Kaposi’s sarcoma-associated herpesvirus lytic replication. The review is well written with good Figures that explain several important aspects of Kaposi Sarcoma lytic replication. In the ribosomal manipulation section, information about the role of HIF2alpha in the translation initiation of viral proteins is missing.

Immunofluorescence figures are very informative; perhaps showing similar results in iSLK-derived cells would strengthen the review.

Minor changes:

-Figure 1. Labeling for “A” and “B” is missing. The reactivation procedure should be explained in detail. Doxycycline alone?

Author Response

Reviewer 3

In the ribosomal manipulation section, information about the role of HIF2alpha in the translation initiation of viral proteins is missing.

This has now been added.

Immunofluorescence figures are very informative; perhaps showing similar results in iSLK-derived cells would strengthen the review.

Unfortunately, we have only done these experiments in TREX cells. Therefore, where possible we have added references in the figure legends, detailing similar findings previously reported in iSLKs, for example in Figure 2 and Figure 3.

Minor changes:

-Figure 1. Labeling for “A” and “B” is missing. The reactivation procedure should be explained in detail. Doxycycline alone?

Labelled corrected and figure legends modified.

Round 2

Reviewer 2 Report

Comments and Suggestions for Authors

The review has been significantly improved, and the authors have addressed my concerns.

A new, minor comment is in their new discussion of lytic contribution to oncogenesis - they might discuss vIL6 and its role (recently reviewed here: https://pubmed.ncbi.nlm.nih.gov/39772207/).

Author Response

Reviewer 2

The review has been significantly improved, and the authors have addressed my concerns.

A new, minor comment is in their new discussion of lytic contribution to oncogenesis - they might discuss vIL6 and its role (recently reviewed here: https://pubmed.ncbi.nlm.nih.gov/39772207/).

Apologies for the omission, we have now briefly discussed the role of vIL-6 in KSHV-mediated tumourigenesis.